# Full spectrum flow cytometry reveals mesenchymal heterogeneity in first trimester placentae and phenotypic convergence in culture, providing insight into the origins of placental mesenchymal stromal cells

Anna Leabourn Boss[1,2]*, Tanvi Damani[2], Tayla J Wickman[1], Larry W Chamley[1], Joanna L James[1†], Anna ES Brooks[2,3†]

[1]Department of Obstetrics and Gynaecology, Faculty of Medical and Health Sciences, University of Auckland, Auckland, New Zealand; [2]School of Biological Sciences, University of Auckland, Auckland, New Zealand; [3]Maurice Wilkins Centre, University of Auckland, Auckland, New Zealand

**\*For correspondence:**
a.boss@auckland.ac.nz

†These authors contributed equally to this work

**Competing interest:** The authors declare that no competing interests exist.

**Abstract** Single-cell technologies (RNA-sequencing, flow cytometry) are critical tools to reveal how cell heterogeneity impacts developmental pathways. The placenta is a fetal exchange organ, containing a heterogeneous mix of mesenchymal cells (fibroblasts, myofibroblasts, perivascular, and progenitor cells). Placental mesenchymal stromal cells (pMSC) are also routinely isolated, for therapeutic and research purposes. However, our understanding of the diverse phenotypes of placental mesenchymal lineages, and their relationships remain unclear. We designed a 23-colour flow cytometry panel to assess mesenchymal heterogeneity in first-trimester human placentae. Four distinct mesenchymal subsets were identified; CD73+CD90+ mesenchymal cells, CD146+CD271+ perivascular cells, podoplanin+CD36+ stromal cells, and CD26+CD90+ myofibroblasts. CD73+CD90+ and podoplanin + CD36+ cells expressed markers consistent with cultured pMSCs, and were explored further. Despite their distinct ex-vivo phenotype, in culture CD73+CD90+ cells and podoplanin+CD36+ cells underwent phenotypic convergence, losing CD271 or CD36 expression respectively, and homogenously exhibiting a basic MSC phenotype (CD73+CD90+CD31-CD144-CD45-). However, some markers (CD26, CD146) were not impacted, or differentially impacted by culture in different populations. Comparisons of cultured phenotypes to pMSCs further suggested cultured pMSCs originate from podoplanin+CD36+ cells. This highlights the importance of detailed cell phenotyping to optimise therapeutic capacity, and ensure use of relevant cells in functional assays.

## Editor's evaluation

Placental mesenchymal stromal cells (pMSCs) are of interest in therapeutic applications. These cells are typically generated by culturing cells isolated from the villous core of the placenta. However, the villous core is comprised of multiple cell types and the heterogeneity of these cells is often not considered. Consequently, the origin of pMSCs under commonly used culture conditions remains unclear. In the present study the authors have used sophisticated flow cytometry analysis to characterize the heterogeneity of subtypes in the placental villus core in the first trimester of gestation and present convincing data that a specific subpopulation identified likely corresponds to pMSCs in

culture generated using standard isolation protocols. This study will be valuable to scientists investigating the use of placental mesenchymal stromal cells in a therapeutic context.

## Introduction

Mesenchymal stromal cells (MSCs) are a heterogeneous population of cells with secretory, immuno-modulatory, and homing properties (*Viswanathan et al., 2019*). This population encompasses fibroblasts, myofibroblasts, pericytes, and tissue-specific progenitor populations. Traditionally researchers defined in vitro expanded MSCs by (1) plastic adherence, (2) expression of CD73, CD90, and CD105, lack of hematopoietic and endothelial marker expression (CD31, CD34, CD45, HLA-DR), and (3) the capacity to differentiate into adipocyte, chondrocyte and osteoblast lineages in vitro (*Dominici et al., 2006*). However, despite MSCs from different tissue sources (i.e. bone-marrow, adipose, umbilical cord, placenta) sharing expression of the minimal criteria antigens after in vitro culture, they functionally differ in a tissue-specific manner (*Du et al., 2016b*; *Kozlowska et al., 2019*). This has been more recently acknowledged in a refined ISCT position statement that requires the tissue origin of MSCs to be provided along with evidence for their in vitro and in vivo functionality (*Viswanathan et al., 2019*).

It has become clear that in vitro culture may have significant impact on MSC phenotype, meaning that cultured MSCs may have a different phenotype from their in vivo/freshly isolated counterparts. For example, prior to culture adipose-derived MSCs are CD34+, whilst in vitro culture induces the loss of CD34 expression (*Lin et al., 2012*). The MSC-associated marker CD271 is also lost with plastic culture (*Brooks et al., 2020*), and MSC αSMA expression can be heavily impacted by culture media (*Boss et al., 2020*). Thus, the culture conditions used to expand MSCs can mask their true in vivo phenotype and heterogeneity. Furthermore, in any set of culture conditions the most competitive stromal cells will thrive and take over the culture, whilst other functionally diverse cells may be diluted out or lost. Understanding the true in vivo phenotype, and optimising culture conditions to minimize phenotypic 'drift' in vitro may enable researchers to better take advantage of the tissue specific properties of MSCs, and enable more accurate study of their roles in vivo.

Placental MSCs (pMSCs) are routinely isolated from the core of placental villi and propagated in culture (*Abumaree et al., 2013*; *Castrechini et al., 2010*; *Kusuma et al., 2018*). The placenta is a highly vascularised organ that develops rapidly over gestation in order to act as the conduit for nutrient and gas exchange between mother and fetus. pMSCs have been reported to reside in a perivascular niche where they can influence vascular development and function (*Castrechini et al., 2010*). In line with this, cultured pMSCs have well established angiogenic and haematopoietic paracrine effects in vitro and in vivo (*Chen et al., 2009*; *Du et al., 2016a*). pMSCs are typically isolated by gross tissue culture methods (explant outgrowths or enzymatic digestion followed by plastic adherence), rather than by expression of specific cell surface markers (*Abumaree et al., 2013*; *Pelekanos et al., 2016*). However, using gross digestion techniques it is impossible to determine the niche in which pMSCs reside. Previous work has assumed that in vitro pMSC expression of CD146, a marker with established perivascular expression in vivo, provides evidence of perivascular origins (*Castrechini et al., 2010*). However, work with bone marrow MSCs suggest CD146 expression is a tissue culture artefact (*Blocki et al., 2013*), and chipcytometry has localised in vitro MSC markers CD73, CD105 and CD90 outside of the perivascular niche in term placentae (*Consentius et al., 2018*). Together this sheds doubt on the perivascular origins of pMSCs, and highlights the need to better characterise their ex vivo phenotype in order to understand their functional role in vivo.

Our understanding of mesenchymal heterogeneity is expanding exponentially, and advances in single-cell technologies have uncovered a wide spectrum of mesenchymal and endothelial phenotypes with specific functions and spatial distributions (*Suryawanshi et al., 2018*; *Takeda et al., 2019*; *Vijay et al., 2019*). Mesenchymal heterogeneity can be used to spatially localise cells, identify cells involved in morphogenesis, angiogenesis or quiescence, and characterise inflammatory, anti-inflammatory, or invasive phenotypes (*Cimini and Kishore, 2021*; *Mezawa et al., 2019*; *Mezheyeuski et al., 2020*; *Nazari et al., 2016*; *Quintanilla et al., 2019*). Dramatic advances in flow cytometry over the past few decades have enabled the simultaneous quantification of upwards of 30 antigens, allowing detection of in depth cell heterogeneity. Whilst initiated in the haematopoietic field, the wealth of data produced using multicolour flow cytometry has driven the use of this technique for analysis of other cell types, including the mesenchymal lineages. To date the largest panel used to assess mesenchymal

cells (16-colours) was designed for analysis with conventional flow cytometry (*Brooks et al., 2020*). However, spectral flow cytometry has significant potential to further improve analysis of mesenchymal cells. The ability of spectral flow cytometry to measure a larger portion of the spectral signature of each fluorophore, and thus enable small differences between fluorophores to be discriminated, combined with the ability to remove autofluorescence (a prominent feature of mesenchymal cells) paves the way for assessing cells previously regarded as challenging for multicolour flow cytometry.

Here, we developed what to our knowledge is the largest multicolour spectral flow cytometry panel employed to date, in order to characterise the mesenchymal heterogeneity in first trimester placental tissue, and to ascertain the impact of in vitro MSC culture conditions on the subsequent phenotype of these populations.

## Results

### Design of a 23-colour flow cytometry panel to characterise the placental villous core

In order to investigate mesenchymal populations of the placental core a 23-colour flow cytometry panel (Panel One) was developed by starting from a 16-colour panel designed to investigate adipose-derived stromal vascular fraction on a conventional flow cytometer (BD FACS Aria II) (*Brooks et al., 2020*). The antigens included in this panel are based on the current literature surrounding the placenta, pMSCs, stromal/mesenchymal populations, endothelial progenitors and haematopoietic cells (*Table 1*).

Placental explants were dissected from the villous tissue and enzymatically digested to obtain a single cell suspension for flow cytometry (Figure 6). To enable us to focus on the mesenchymal core cells of interest, we needed to exclude trophoblasts (epithelial cells found in a bilayer on the outer edge of placental villi), endothelial cells, and hematopoietic cells. Whilst the initial enzymatic digestion removed the outer syncytiotrophoblast layer, inclusion of the cytotrophoblast maker β4 integrin in Panel One ensured any potential contaminating cytotrophoblasts directly beneath this were also excluded from analysis (*Table 1*, *Figure 1B and C*). Endothelial cells were identified by expression of CD31 and CD144, specific expression of which is seen in blood vessels and early endothelial cell cords in first trimester placental villi (*Figure 1E*). The hematopoietic marker CD45 was used to exclude all cells of this lineage, while CD235a was used to identify and exclude red blood cells (*Table 1*). All remaining cells were considered to constitute the stromal fraction that makes up the mesenchymal core of the villi.

### Full spectrum flow cytometry uncovers heterogeneous mesenchymal populations in the core of first trimester placental villi

First trimester placental villous explants were enzymatically digested to obtain a single-cell suspension that was stained with a master mix for Panel One and analysed on a Cytek Aurora. Due to the complexity of data obtained from a 23 colour panel, unbiased high-dimensional clustering of the data was undertaken using viSNE (Cytobank). viSNE enables visualization of high-dimensional single-cell data and is based on the t-Distributed Stochastic Neighbour Embedding (t-SNE) algorithm (*Amir et al., 2013*). To run the algorithm, debris, doublets, dead cells, β4 integrin +cytotrophoblasts and CD45+/CD235a hematopoietic cells were excluded by manual gating, then an equal number of cells from each villous core digest was selected using down-sampling (n=5) (*Figure 2*). Combining viSNE with known population marker expression, five distinct subsets were uncovered (*Figure 2*). Endothelial cells co-expressed CD31 and CD34 (Subset One) and made up 3.9% ± 2.5% of villous core cells (*Figure 3C and D*). CD73$^+$CD90$^+$ mesenchymal cells (Subset Two) made up 6.8% ± 9.7% of villous core cells on average, but decreased in abundance with increasing gestational age within the first trimester (*Figure 2C*). This gestational difference was confirmed by the inclusion of data from 22 additional first trimester villous core digests, analysed using a smaller panel targeted at cell sorting but containing the markers required to identify this population (*Figure 2D*). Together this data showed that at <10 weeks of gestation CD73$^+$CD90$^+$ cells constituted 14.02+8.84% of villous core cells, but significantly decreased to 0.77+0.83% of villous core cells at ≥10 weeks of gestation (*P*<0.001, n=27, placentae = 7–13.1 weeks of gestation) (*Figure 3C*). Perivascular cells (Subset Three) were identified by their expression of CD146$^+$CD271$^+$, and constituted 10.3% ± 4.7% of villous core cells (*Figure 3C*).

**Table 1.** Antigens included in Panel One and their functional roles with respect to the villous core. Antigens are grouped by cell type (grey and white boxes).

| Antigen | Full name/s | Functional capacity/relevance | References |
|---|---|---|---|
| ITGβ4/CD104 | β4 integrin | Cell adhesion molecule that uniquely identifies cytotrophoblasts in the placenta. | *James et al., 2015* |
| CD45 | Protein tyrosine phosphatase, receptor type, C. lymphocyte common antigen | Expressed on all nucleated haematopoietic cells. | *Hermiston et al., 2003* |
| HLA-DR | Human Leukocyte Antigen – DR isotype | Expressed by antigen presenting cells i.e. macrophages, MHC class II cell surface receptor, involved in antigen presentation and adaptive immunity, | *Cruz-Tapias et al., 2013* |
| CD235a/GPA | Glycophorin A | Red blood cell marker, identifies early emerging RBCs/erythroblasts from haemogenic endothelium | *Garcia-Alegria et al., 2018*; *Mao et al., 2016* |
| CD41 | Integrin alpha-IIb | Platelet marker. Expressed by earliest emerging haematopoietic cells from haemogenic endothelium | *Garcia-Alegria et al., 2018*; *Li et al., 2005* |
| CD117/cKIT | Mast/stem cell growth factor receptor | Receptor expressed on haematopoietic stem cells and involved in their differentiation | *Rönnstrand, 2004* |
| CD144 /VE-cadherin | Vascular endothelial cadherin | Endothelial cell-cell adherens junctional marker, stabilises vessels, inhibits vascular growth, regulates vascular permeability. | *Giannotta et al., 2013* |
| CD34 | Haematopoietic Progenitor Cell Antigen | Expressed by haematopoietic and vascular progenitors and adipose-derived MSCs. Transmembrane phosphoglycoprotein, thought to identify early placental progenitors | *Brooks et al., 2020*; *Sidney et al., 2014*; *Yoder, 2009* |
| CD31/PECAM1 | Platelet endothelial cell adhesion molecule | Expressed by endothelial cells, platelets, macrophages and Kupffer cells, granulocytes, lymphocytes, megakaryocytes. Adhesion molecule found at endothelial intercellular junctions. | *Marelli-Berg et al., 2013* |
| VEGFR2/KDR | Vascular Endothelial Growth Factor Receptor 2/Kinase Insert Domain Receptor | Expressed by endothelial cells and thought to identify placental core progenitors. Receptor for angiogenic VEGF, involved in vasculogenesis and angiogenesis. | *Demir et al., 2007* |
| CD54/ICAM1 | Intercellular Adhesion Molecule 1 | Expressed at low levels on endothelial cells, monocytes and lymphocytes. Increased expression in response to inflammatory cytokines. | *Hubbard and Rothlein, 2000* |
| CD36/FAT | Platelet glycoprotein 4/ fatty acid translocase | Expressed by microvascular endothelial cells, and fibroblasts. Has an anti-angiogenic effect via binding Thrombospondin 1. Involved in fatty acid uptake. | *Dye et al., 2001*; *Heinzelmann et al., 2018*; *Silverstein and Febbraio, 2009* |
| CD90 /Thy-1 | Thymocyte differentiation antigen 1 | Expressed by MSCs, haematopoietic stem cells, fibroblasts, myofibroblasts. | *Viswanathan et al., 2019* |
| CD73 /NT5E | Ecto-5'-nucleotidase | Expressed by MSCs and endothelial cells. Works with CD39 to convert extracellular ATP to adenosine to create immunosuppressive effect. | *Roh et al., 2020*; *Viswanathan et al., 2019* |
| CD39 /NTPDase | Ectonucleoside triphosphate diphosphohydrolase-1 | Upregulated by MSCs to suppress lymphocyte activation. Immunosuppressive actions via the conversion of extracellular ATP (inflammatory) into adenosine (anti-inflammatory) | *Saldanha-Araujo et al., 2011*; *Zhao et al., 2017* |

*Table 1 continued on next page*

*Table 1 continued*

| Antigen | Full name/s | Functional capacity/relevance | References |
|---|---|---|---|
| CD55/DAF | Decay-accelerating factor | Expressed by MSCs. Complement regulatory protein, inhibits C3 convertases thereby creating a threshold for complement activation, increased expression correlated with evasion of innate immune system | *Ruiz-Argüelles and Llorente, 2007*; *Soland et al., 2013* |
| CD271 /NGFR / p75NTR | Low-affinity Nerve Growth Factor Receptor | Used in MSC and pericyte identification. Hypothesised to identify "stem-cell" or progenitor populations with superior differentiation and colony forming capacity. | *Barilani et al., 2018*; *Kumar et al., 2017* |
| CD146/MCAM | Melanoma cell adhesion molecule | Adhesion molecule expressed by pericytes, endothelial cells and smooth muscle cells. Involved in the regulation of angiogenesis and vessel permeability. | *Crisan et al., 2009*; *Leroyer et al., 2019* |
| CD248 | Endosialin/ tumor endothelial marker 1 | Pericyte and stromal cell marker. Involved in cell–cell adhesion, and host defence. | *Lax et al., 2010*; *Tomkowicz et al., 2010* |
| CD142/TF | Tissue Factor/ thromboplastin | Expression correlated with pericytes, smooth muscle and fibroblasts. Activates blood clotting after injury, located outside the vasculature, | *Abe et al., 1999*; *Morrissey, 2004* |
| CD26/DDP4 | Dipeptidyl peptidase-4, adenosine deaminase complexing protein 2 | Expressed by many tissues; T-cells, epithelial cells, ESCs, progenitor cells, placental myofibroblasts. Serine protease that cleaves a range of chemokines. Downregulation is correlated with increased stromal/myofibroblast proliferation. | *Kohnen et al., 1996*; *Mezawa et al., 2019*; *Ou et al., 2013* |
| PDPN | Podoplanin | Lymphatic vascular marker. Expression is correlated with increased fibroblast migration. Binds to CLEC-2 receptor on platelets. | *Astarita et al., 2012*; *Suchanski et al., 2017* |

Subset Four made up the largest proportion of mesenchymal cells in the core (47.8+17.0%) and were identified by co-expression of podoplanin$^+$CD36$^+$ (*Figure 3C*). The remaining cells, expressing only the myofibroblast-associated markers CD26$^+$CD90$^+$, were classified as Subset Five and constituted 31.3% ± 12.6% of villous core cells. Subset Five were negative for all other markers investigated, suggesting they are more differentiated myofibroblast-like cells (*Figure 3C*).

## Mesenchymal subsets have distinct phenotypes indicating functionally diverse roles in the villus core

In order to consider what the different subsets functionally represent within the core of placental villi, we further interrogated their phenotype and proportional contribution to the villus core, with consideration to which subset phenotypes pMSCs may originate from. CD73$^+$CD90$^+$ cells (Subset Two) were the only villous population to co-express CD90 and CD73, both MSC-associated markers. This subset also expressed podoplanin (correlated with increased migratory capacity), low levels of CD142 (involved in blood coagulation [*Morrissey, 2004*]), and had heterogeneous expression of CD146 and CD271 (perivascular associated markers) (*Figure 3B*). A lack of CD26 (associated with myofibroblasts and cell proliferation) made this population distinct from the other perivascular and mesenchymal cell populations identified in the villous core using Panel One.

Perivascular cells (Subset Three), identified by expression of CD271 or CD146, did not co-express CD73, CD142 or podoplanin. However, they did express CD26 and CD90, and in line with this they co-localised with Subset Five (also CD26$^+$CD90$^+$) on the viSNE plot (*Figure 3B*). The phenotype of CD271$^+$CD146$^+$ cells was relatively conserved between placentae with some inter-placental variance in CD271, CD36 and CD90 expression (*Figure 3B*).

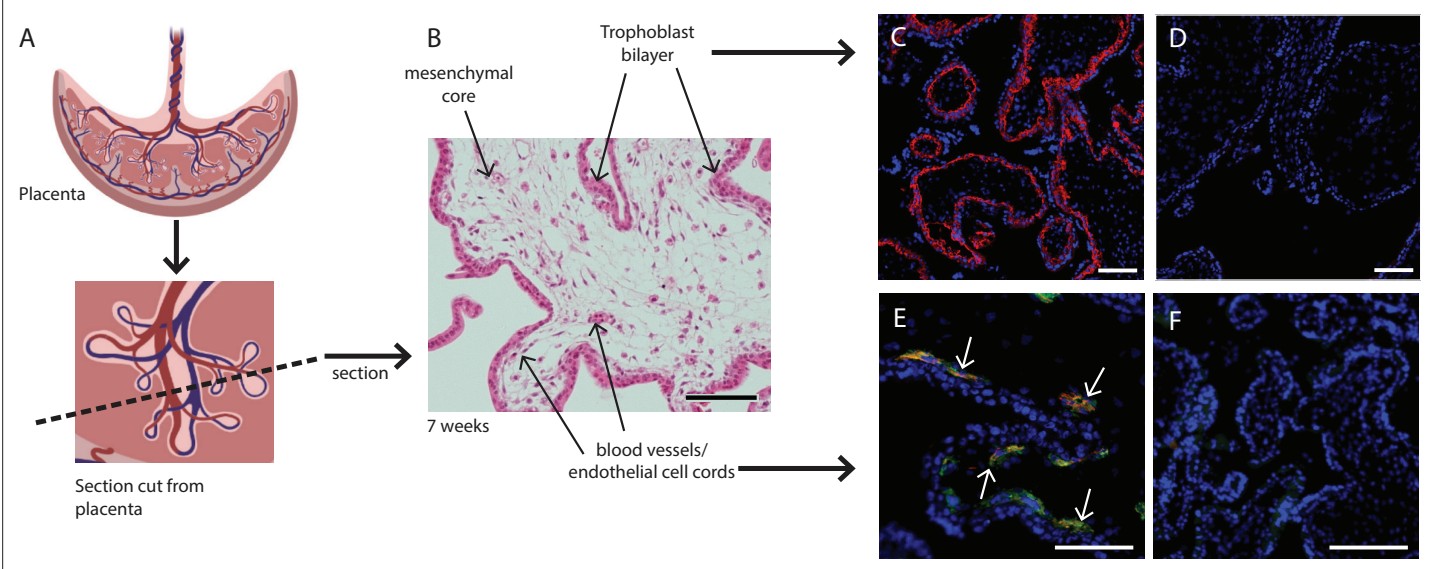

**Figure 1.** Placental villus structure and specificity of markers used to exclude unwanted cell populations. (**A**) Placental villous morphology and plane of section, (**B**) haematoxylin and eosin staining of a thin section though a placental villus (7.1 weeks), localisation of (C) β4 integrin (red) to cytotrophoblasts and (**E**) CD144 (red) and CD31 (green) to blood vessels (white arrows) in placental villus sections confirmed antibody specificity. No fluorescence is seen in negative IgG controls (**D, F**) run simultaneously. Nuclei are counterstained with DAPI (blue). Scale bar = 100 μm. Rendered images in (A) have been acquired from Biorender.com.

Subset Four was identified by expression of podoplanin and CD36, and co-expressed CD26, CD90, and CD142, but was negative for the MSC-associated marker CD73 (*Figure 3B*). As the largest population within the villous core and expressing proliferative/migratory associated markers (CD26, podoplanin [*Suchanski et al., 2017*]) this subset was considered the most likely in vivo candidate to contribute to pMSCs that are derived in explant derived cultures.

## Phenotypically divergent mesenchymal stromal cell populations converge after in vitro culture

In order to investigate the origins of in vitro cultured pMSCs in relation to the subsets identified with Panel One we used fluorescence activated cell sorting (FACS) to isolate subsets of interest from first trimester villous digests and assessed their phenotype in relation to cultured pMSCs obtained using an explant outgrowth method. Here we focussed on two ex vivo cell populations; CD73⁺CD90⁺ cells (Subset Two) as the only population to phenotypically align with the ISCT marker criteria for MSCs (*Dominici et al., 2006*), and podoplanin⁺CD36⁺ cells (Subset Four), which expressed markers of proliferative/migratory cells (characteristics associated with pMSCs). Following sorting, cells were cultured for 7 days in EGM-2 previously demonstrated to sustain fetal-derived pMSCs (*Boss et al., 2020*; *Figure 4*). In culture, both populations underwent phenotypic convergence, forming cells with a mesenchymal-like morphology, that were CD90⁺CD73⁺CD142⁺CD271⁻CD36⁻ at day 7 of culture (*Figure 4B and C* and *Figure 4—figure supplement 1*).

Despite this overall convergence, differences between the two populations did remain. The proliferation/myofibroblast associated marker CD26 was only expressed by a small proportion of CD73⁺CD90⁺ cells at the time of isolation and although the percentage of CD26⁺ cells increased over time in culture, most cells remained negative suggesting this marker was not upregulated with the culture conditions used here. Conversely, all podoplanin⁺CD36⁺ cells were CD26⁺ at isolation and remained CD26⁺ after culture. Although CD146, was used to identify perivascular cells in the placenta (*Castrechini et al., 2010*), was not assessed on cells during isolation (this marker was not contained in Panel Two), phenotyping experiments with Panel One demonstrated that a subpopulation of CD73⁺CD90⁺ cells expressed CD146 (5.89+5.82%) and therefore, could be perivascular. Whereas, podoplanin⁺CD36⁺ cells were negative for CD146. After 7 days culture this CD146⁺ subpopulation was still present in CD73⁺CD90⁺ cells (15.55+6.87%). However, all podoplanin⁺CD36⁺ cells expressed

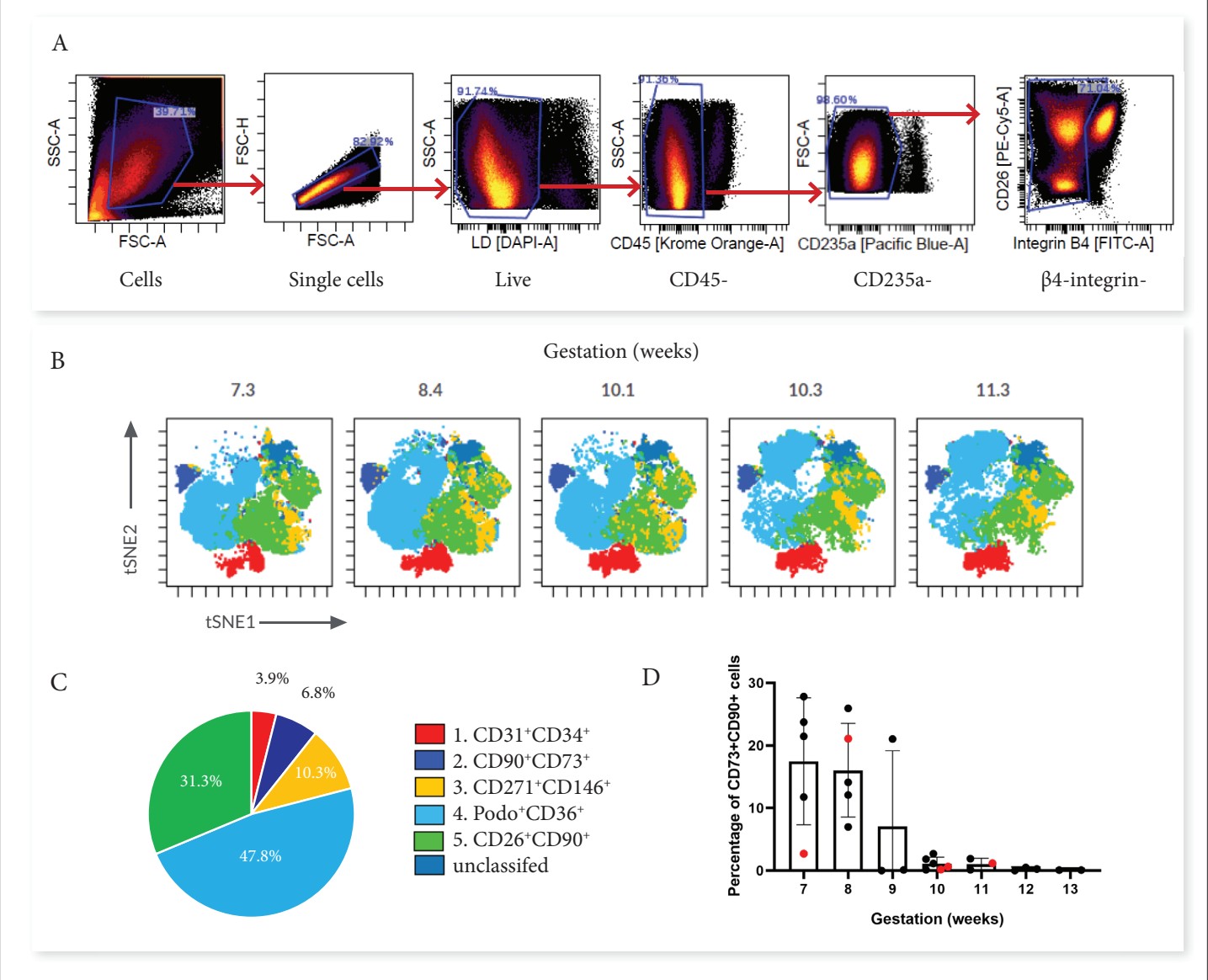

**Figure 2.** Categorisation of placental villus core subsets using Panel One markers. (**A**) Samples were gated to exclude debris, doublets, dead cells, hematopoietic cells (CD45$^+$ and CD235a$^+$), and cytotrophoblasts (β4 integrin). (**B**) Marker expression was used to categorise five subsets that were overlaid onto viSNE plots generated in Cytobank. (**C**) The average percentage contribution of each subset is presented as a pie chart. (**D**) A scatter plot with bars depicting the mean percentage of CD73$^+$CD90$^+$ cells from villous core cells across first trimester analysed on an Aria II (n=24, black) or an Aurora spectral analyser (n=5, red). Error bars represent the standard deviation of the mean.

CD146 after seven days in culture, demonstrating differential response of these subsets to the in vitro culture conditions (*Figure 4D and E*, and *Figure 4—figure supplements 1 and 2*).

In order to determine whether cultured pMSCs are likely to be made up of CD73$^+$CD90$^+$ or podoplanin$^+$CD36$^+$ populations, we simultaneously characterised explant isolated pMSCs that had been generated and cultured in the same medium (EGM-2) (n=3, p2-6). This demonstrated that pMSCs were CD90$^{low/neg}$CD73$^+$CD26$^+$CD142$^+$CD146$^+$podo$^+$CD34$^-$CD271$^-$CD36$^-$(*Figure 4F* and *Figure 4—figure supplements 1 and 2*), more closely aligning with the phenotype of cultured podoplanin$^+$CD36$^+$ cells (Subset Four).

In previous work we have demonstrated that pMSCs upregulate the contractile markers αSMA and calponin when transferred from EGM-2 to standard MSC medium (advanced-DMEM/F12) (*Boss et al., 2020*). In order to investigate whether CD73$^+$CD90$^+$ cells (Subset Two) or podoplanin$^+$CD36$^+$ cells (Subset Four) could similarly upregulate contractile markers, after 7 days in EGM-2 the medium was changed

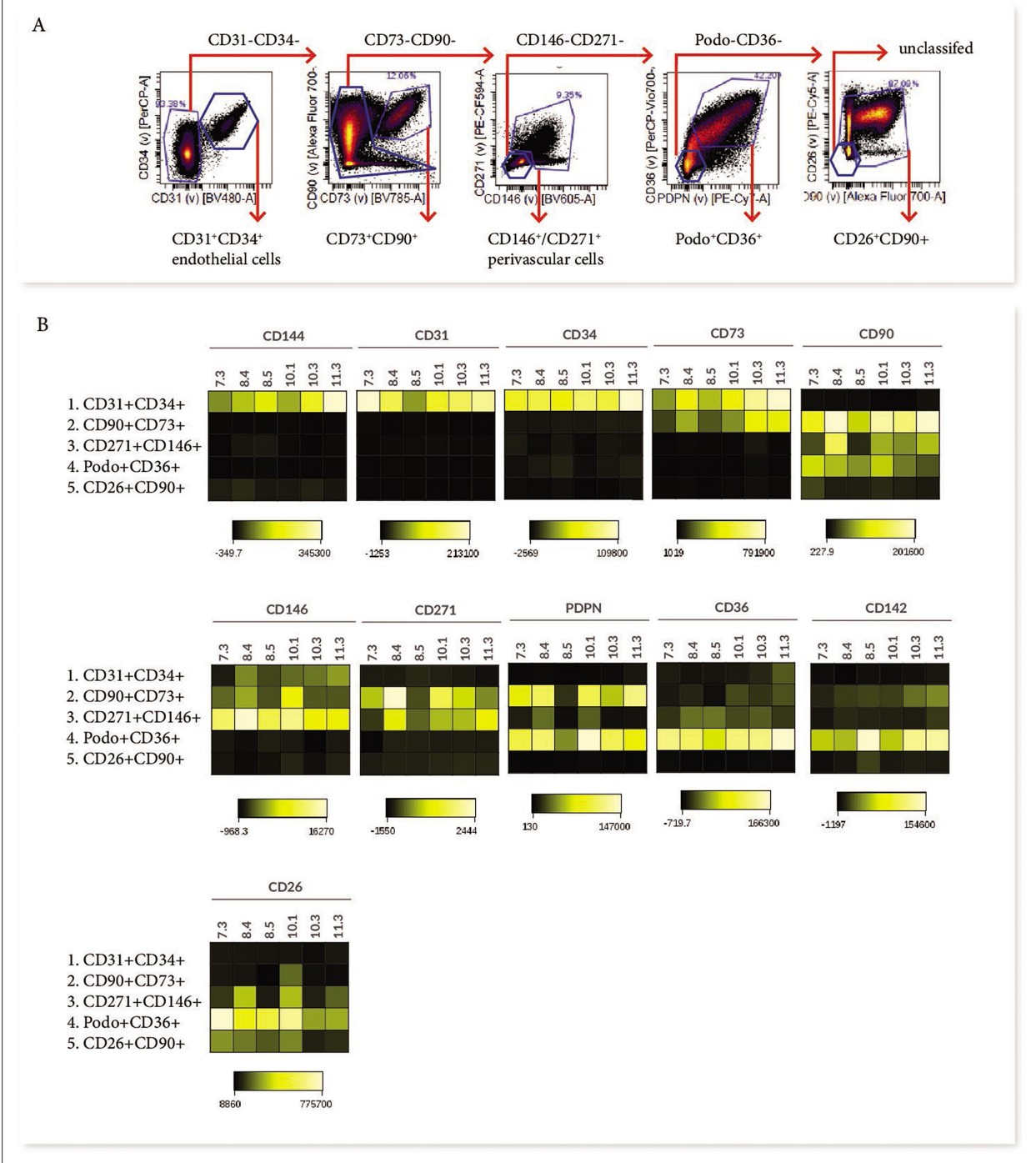

**Figure 3.** Phenotypic characterisation of villous core subsets. (**A**) The gating strategy used to identify subsets (CD31+CD34+, CD73+CD90+, perivascular cells, podoplanin+CD36+ and CD26+CD90+), and (B) heat maps comparing the expression of specific antigens between subsets (n=5).

to advanced-DMEM/F12. Whilst in EGM-2 very few cells in either population expressed αSMA and no cells expressed calponin, in advanced-DMEM/F12, both populations upregulated both of these markers (*Figure 5*). Expression of the more mature smooth muscle marker MYH-11 was not observed in either CD73+CD90+ or podoplanin+CD36+ cells cultured in either EGM-2 or advanced DMEM/F12 (*Figure 5*).

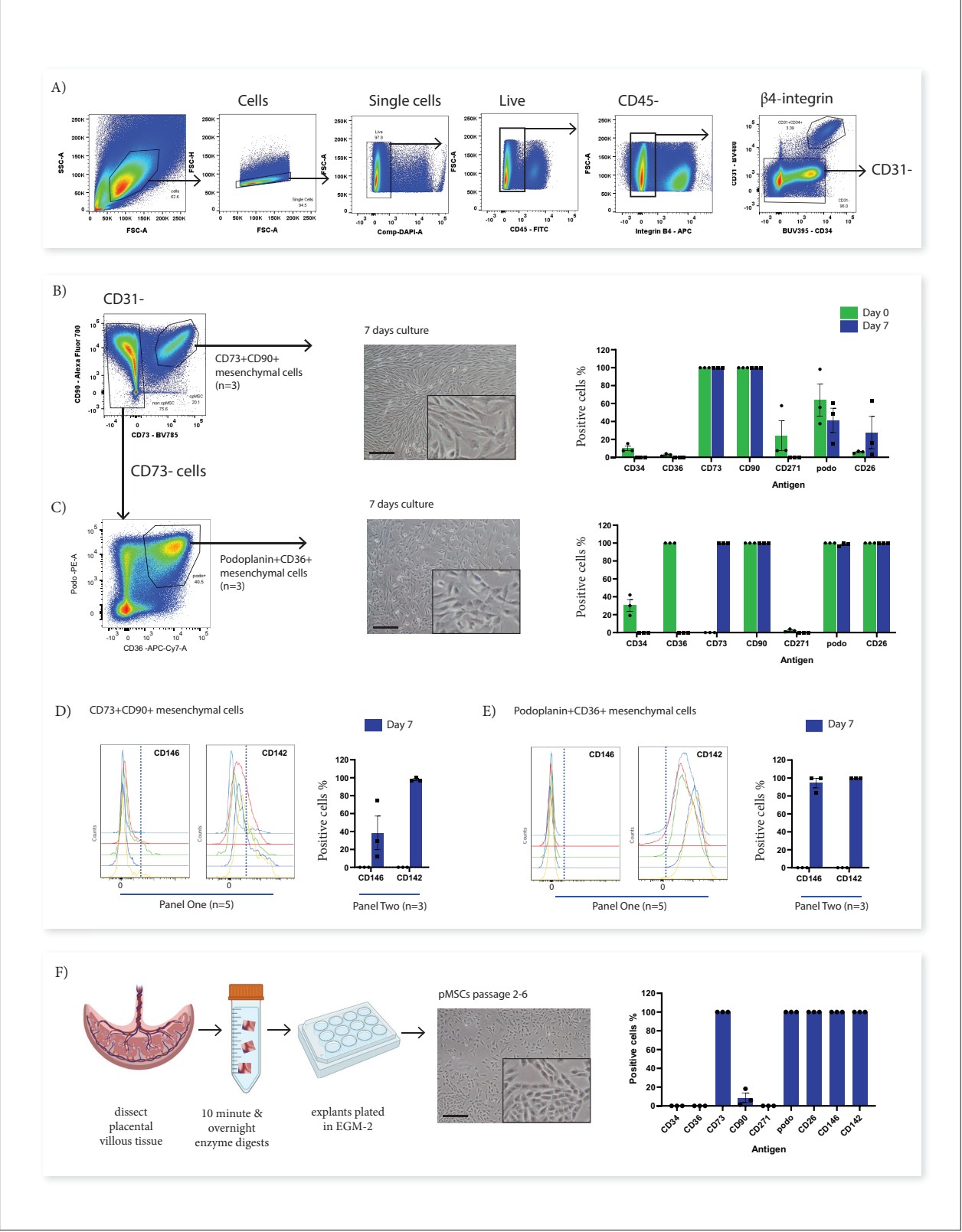

**Figure 4.** In vitro culture of CD73+CD90+ and podoplanin+CD36+ cells. (**A**) FACS sorting was used to isolate CD73+CD90+ and podoplanin+CD36+ cells from placental villous core cells (n=3). (**B**) Morphology and phenotype of CD73+CD90+ cells after 7 days in culture. (**C**) Phenotype of podoplanin+CD36+ cells after 7 days in culture. (**D**) CD146 and CD142 expression on CD73+CD90+ analysed with Panel One (day 0, n=5) or at 7 days after culture (n=3).

*Figure 4 continued on next page*

*Figure 4 continued*

(**E**) CD146 and CD142 expression on podoplanin⁺CD36⁺ cells analysed with Panel One (day 0, n=5) or at 7 days after culture (n=3). (**F**) Isolation of explant-derived pMSCs, and morphology and phenotype of passaged pMSCs (n=3). Error bars = standard error of the mean and scale bar = 100 µm.

The online version of this article includes the following figure supplement(s) for figure 4:

**Figure supplement 1.** Representative 2-dimensional flow cytometry plots displaying the phenotype of FACS sorted CD73⁺CD90⁺ and podoplanin⁺CD36⁺ cells after 7 days culture in vitro (n=3), and explant isolated pMSCs after culture in vitro (n=3, p2-6).

**Figure supplement 2.** Flow cytometry histograms displaying the phenotype of FACS sorted CD73⁺CD90⁺ and podoplanin⁺CD36⁺ cells after 7 days culture in vitro (n=3), and explant isolated pMSCs after culture in vitro (n=3, p2-6).

## Discussion

Common sources of MSCs used in clinical trials include; uncharacterised adipose, bone-marrow, and placental stroma. Identifying functionally different mesenchymal cells and defining the origins for tissue-specific MSCs could improve these applications. Here, we used a high-dimensional flow cytometry panel to uncover at least four different mesenchymal populations with distinct expression profiles in the villous core of first trimester placentae, indicating the presence of functionally specialised cells. This mesenchymal heterogeneity could translate to different physiological capabilities important for clinical therapies and aid understanding of the role of these cells in vivo.

Characterising the villous core populations allows us to better understand how the placenta functions and develops, and enables targeting of populations for functional analysis. From the four mesenchymal populations classified in this work three populations, perivascular cells (Subset Three), podoplanin⁺CD36⁺ cells (Subset Four), and CD26⁺CD90⁺ myofibroblasts (Subset Five) all closely aligned with populations described in prior single-cell RNA sequencing data from first trimester placentae (*Suryawanshi et al., 2018*). However, this single-cell RNA sequencing dataset did not report a population that aligned with CD73⁺CD90⁺ cells (Subset Two), instead reporting that only endothelial cells expressed CD73 in the placental core. The ability to assess larger cell numbers by flow cytometry in our work, and the low abundance of CD73⁺CD90⁺ cells after 9 weeks of gestation, means that this population may not have been detected as a distinct population by single cell RNA-seq. Furthermore, RNA expression does not always accurately reflect protein expression (*Reimegård et al., 2021*). This highlights the importance of protein level characterisation of cells from fresh tissue, and the ability of multicolour flow cytometry to undertake this at single cell resolution complements and builds on single cell gene expression datasets.

The first major aspect of this work to consider is what each subset could represent and what its phenotype reflects. CD73⁺CD90⁺ cells were unique from the other mesenchymal populations and could be identified by their expression of CD73, which was shared only with endothelial cells. The abundance of this population at <10 weeks of gestation hints at their involvement in processes that occur early in gestation such as vasculogenesis or haematopoiesis. CD73⁺CD90⁺ cells co-expressed podoplanin, but were distinct from other mesenchymal subsets in that they did not express CD26. This combined expression pattern alludes to two potential functional roles of these cells in the placenta. First, expression of podoplanin and low expression of CD26 are both associated with a migratory/invasive phenotype (*Mezawa et al., 2019*; *Ward et al., 2019*). As the decline in abundance of these cells after 9 weeks of gestation corresponds to the regression of villi and blood vessels from the distal side of the gestational sac (*Burton and Jauniaux, 2018*), this raises the possibility that CD73⁺CD90⁺ cells could be involved in this regression of the villi, or they arise transiently as the result of an early endothelial to mesenchymal transition in the placenta whereby, endothelial cells involved in vascular regression lose their junctional markers and gain a mesenchymal/migratory phenotype (*Pinto et al., 2016*). Secondly, expression of podoplanin and a lack of CD26 expression on CD73⁺CD90⁺ cells indicates potential roles in haematopoiesis. Several studies have demonstrated that inhibiting CD26 enhances homing, engraftment and differentiation of hematopoietic stem cells (*Christopherson et al., 2004*; *Kawai et al., 2007*), and thus a lack of CD26 expression on CD73⁺CD90⁺ cells could suggest they are involved in blood lineage migration/differentiation as both Hofbauer cells (placental macrophages) and nucleated red blood cells are found in the villous stroma in the first trimester (*Van Handel et al., 2007*). Furthermore, podoplanin identifies connective stroma in lymphoid organs that produces a network of collagen-rich fibres to enable movement of lymphocytes via interactions with leukocyte CLEC-2 receptors (*Link et al., 2011*; *Nazari et al., 2016*; *Onak Kandemir et al., 2019*;

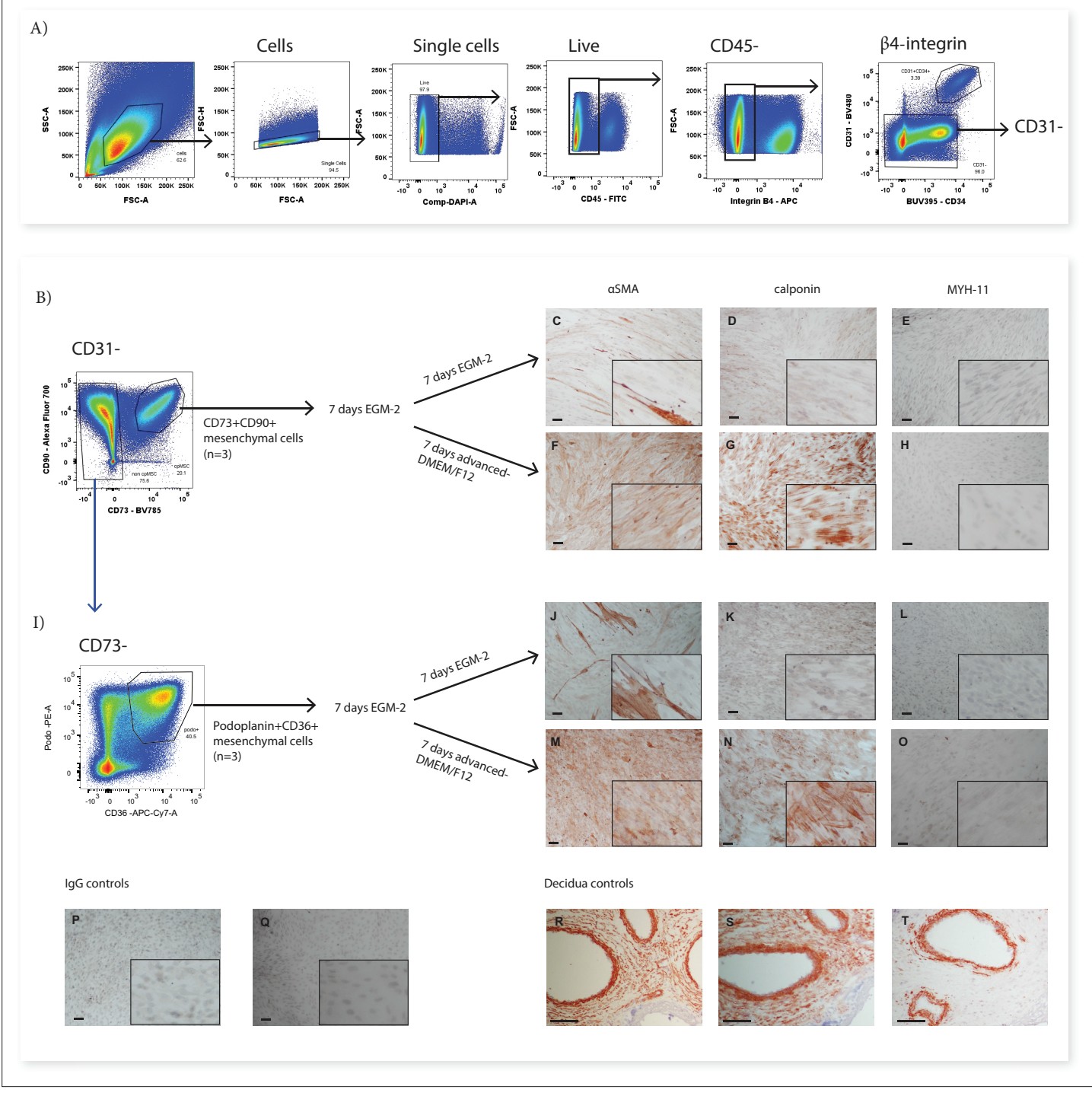

**Figure 5.** CD73$^+$CD90$^+$ and podoplanin$^+$CD36$^+$ cells upregulate markers consistent with contractile cells. (**A**) FACS sorting was used to isolate CD73$^+$CD90$^+$ and podoplanin$^+$CD36$^+$ cells from placental villous core digests (n=3). CD73$^+$CD90$^+$ (**B**) and podoplanin$^+$CD36$^+$ (**I**) cell expression of αSMA (**C, F, J, M**), calponin (**D, G, K, H**) or MYH-11 (**E, H, L, O**) following 7 days of culture in advanced-DMEM/F12 or EGM-2. Irrelevant mouse IgG (**P**) and rabbit IgG (**Q**) were used as negative controls. Decidual sections containing spiral arteries with intact smooth muscle layers were used as positive controls for staining (**R–T**). Scale bar = 100 µm.

*Wang et al., 2011*). The stroma of early placental villi is poorly vascularised therefore podoplanin$^+$ cells could enable blood lineage cells (including Hofbauer cells) to migrate throughout the villous core (*Ingman et al., 2010*). Indeed, Hofbauer cells have been co-localised with podoplanin expressing stroma in term placentae (*Onak Kandemir et al., 2019*). Together this suggests CD73$^+$CD90$^+$ cells

may represent a functionally divergent population, that is abundant in the early first trimester placenta which is specialised to facilitate/promote hematopoietic migration and differentiation, and morphogenesis of the placenta during regression of the distal villi.

The largest subset identified, podoplanin⁺CD36⁺ cells (Subset Four) was hypothesised to constitute the majority of connective villous stroma and play an important role in villous stroma expansion. Similar to CD73⁺CD90⁺ cells, the expression of podoplanin in this population suggests they may play a role in facilitating hematopoietic transport throughout the poorly vascularised first trimester stroma, and may have a migratory phenotype. However, podoplanin⁺CD36⁺ cells had a high expression of CD26 which identifies fibroblasts involved in wound healing/fibrosis and the associated extracellular matrix deposition (*Soare et al., 2020*; *Worthen et al., 2020*) suggesting an alternative or additional role in villous expansion and growth. The corresponding podoplanin⁺CD36⁺ population observed in the prior single cell RNA-seq dataset, expressed genes associated with angiogenesis (*IL6, ANGPTL-4, TNFAIP6*), suggesting they may also provide a paracrine stimulus for placental vascular development which is undergoing vasculogenesis and angiogenesis over first trimester (*Chang et al., 2012*; *Hato et al., 2008*; *Suryawanshi et al., 2018*).

Perivascular cells (Subset Three) were identified by their expression of either CD271 or CD146, both markers that localise in a perivascular niche (*Castrechini et al., 2010*; *Lv et al., 2014*). CD26⁺CD90⁺ myofibroblasts (Subset Five) were negative for all other markers in Panel One and therefore, were hypothesised to represent contractile/less proliferative myofibroblasts. Indeed, prior single-cell RNA sequencing work identified that populations aligning with perivascular cells and CD26⁺CD90⁺ myofibroblasts reported here had a respectively low or high expression of markers associated with contractile myofibroblasts and mature perivascular cells (*ACTA2* (αSMA) and *CNN1* (calponin)) (*Kumar et al., 2017*; *Suryawanshi et al., 2018*). The lower expression of contractile markers in perivascular cells in comparison to CD26⁺CD90⁺ myofibroblasts may initially appear unexpected, but as the first trimester placental vasculature is relatively immature it would not necessarily be expected that perivascular cells had high expression of contractile markers at this stage of gestation (*Zhang et al., 2002*). Interestingly, our in vitro experiments demonstrated that both CD73⁺CD90⁺ cells (Subset Two) and podoplanin⁺CD36⁺ cells (Subset Four) upregulated αSMA and calponin when transferred from EGM-2 to advanced-DMEM/F12, paralleling the upregulation observed in explant-isolated first trimester pMSCs when cultured in this medium (*Boss et al., 2020*). The in vivo niche that CD73⁺CD90⁺ cells and podoplanin⁺CD36⁺ cells reside in could determine whether differentiated cells contribute to perivascular or myofibroblast contractile cells and more specific markers may be required to tease apart these differences. Indeed, in term placentae contractile perivascular cells express both αSMA and calponin, but extravascular αSMA⁺ cells are also present and thought to allow stem villi to contract and expand in volume (*Dellschaft et al., 2020*; *Farley et al., 2004*).

Cultured MSCs are often described as homogeneous. However, there is strong evidence that MSCs upregulate/downregulate markers in response to in vitro culture conditions, and underlying functional and phenotypic heterogeneity could be missed (*Blocki et al., 2013*; *Boss et al., 2020*; *da Silva Meirelles et al., 2015*). In the MSC field, CD73 is thought to be ubiquitously expressed by MSCs (*Dominici et al., 2006*). However, that podoplanin⁺CD36⁺ cells upregulate CD73 expression after culture demonstrates expression of this marker may also be a culture artefact. The phenotypic convergence of different mesenchymal populations was further demonstrated by the loss of CD36 and CD271 expression, from podoplanin⁺CD36⁺ and CD73⁺CD90⁺ cells respectively, additional markers that made these populations distinct ex vivo. However, expression of CD26 was unaffected by culture, and CD146 was differentiably affected by culture in the two populations. This demonstrates that the nuances of culture adaptation are cell-specific as much as they may be media-specific.

Changes to "MSC" phenotype do not necessarily reflect gain/loss in function. For example bone-marrow MSCs isolated by their expression of CD146 stabilize endothelial tube formation in vitro, whereas, CD146⁻ MSCs, that subsequently upregulate CD146 in vitro, are unable to stablise tube formation (*Blocki et al., 2013*). Thus, ex vivo phenotype may be a more accurate indicator of a cells' functional capacity and highlights the importance of characterizing mesenchymal cells prior to culture. The four placental mesenchymal populations identified here are likely to play different roles in the placenta, and therefore have different functional and therapeutic capacities. pMSCs are often acquired by macerating the villous tissue in order to obtain a single cell suspension that is plated in plastic tissue culture flasks (*Papait et al., 2020*; *Pelekanos et al., 2016*). Thus all of the populations

identified in this work could be present within such single cell suspensions. However, the population/s with a competitive advantage in culture, either as a result of the culture conditions employed, or as a result of the isolation process itself (enzymes involved, explant-based, plated suspension, enriched for expression of a marker), are likely to outcompete and become the dominant cell in culture. Indeed, our group has demonstrated that under different media conditions fetal pMSCs rather than maternal cells are enriched in culture in EGM-2 (*Boss et al., 2020*). Characterising the explant isolated pMSCs in more depth in this work demonstrated they express podoplanin, CD26, and CD142, making their origins more likely to be podoplanin$^+$CD36$^+$ (Subset Four) rather than CD73$^+$CD90$^+$ cells (Subset Two), perivascular cells (Subset Three) or CD26$^+$CD90$^+$ myofibroblasts (Subset Five). The heterogeneity identified in this work creates an opportunity to select subsets of these mesenchymal cells more suited to specific uses i.e. with anti-inflammatory/immunomodulatory or angiogenic capacity, and to target media conditions that maintain or transform their in vivo phenotype and desired properties. Furthermore, separately culturing these different subsets will allow us to better to understand their functional contributions to placental development, and whether in vitro culture conditions that promote surface marker convergence also result in functional convergence.

Finally, it is important to consider potential limitations of the use of the cell types identified in therapeutic applications, for example the potential pro-coagulant role of cell surface expression of CD142 (Subset Four) (*Moll et al., 2020*), or the prior association of CD36 expression (seen on Subset Four) with adipose progenitors, raising the possibility that these cells could be prone to differentiating down this pathway in vivo (*Gao et al., 2017*; *Hanschkow et al., 2022*). Conversely, the expression of podoplanin (Subset Two and Four), may indicate an increased migratory capacity, which could potentially enhance migration to sites of therapeutic interest when administered in vivo (*Ward et al., 2019*). Future work to understand the in vivo and ex vivo functional capacity of each subset identified will help elucidate both their role in placental physiology, and inform potential downstream applications of these cells or their counterparts that may be present in term placentae (a more accessible source of placental cells for therapeutic applications).

In conclusion, this work has demonstrated how simplistic post-culture MSC phenotyping will fail to detect the abundant mesenchymal heterogeneity that exists within the placenta. The phenotypic heterogeneity uncovered in first trimester placental mesenchymal cells indicates their potentially diverse functional roles and highlights the crucial need to thoroughly characterise mesenchymal cells that are used in functional assays or in a therapeutic capacity. The success of this multicolour flow cytometry panel in uncovering mesenchymal cell heterogeneity paves the way for other panels assessing mesenchymal and vascularised tissues. Panels like this could be used to assess cellular dysfunction in pathologies such as cancer, cardiovascular disease and infection, as well as to interrogate different tissues in order to better understand cell function. Identifying the pathological phenotype of mesenchymal/vascular cells in pathologies could lead to therapeutic targets. Indeed, in the context of pregnancy, emerging data has suggested that MSC function is altered in the pregnancy complication fetal growth restriction, with potential impacts on placental vascular function (*Boss et al., 2021*; *Gillmore et al., 2022*). Thus, this flow cytometry antibody panel, optimised for placental cells, combined with assessment of different stromal subsets identified in this work, could provide an important tool to better understand normal and abnormal placental development and function across gestation.

**Table 2.** Primary conjugated antibodies used to confirm specificity of cytotrophoblast or endothelial markers.

| Antigen | Fluorophore | Clone | Dilution | Supplier |
|---|---|---|---|---|
| β4 integrin | FITC/APC | 450-9D | 1:200 | Thermofisher |
| CD144 | BV421 | 55–7 H1 | 1:200 | BD |
| CD31 | BV480 | WM59 | 1:200 | BD |

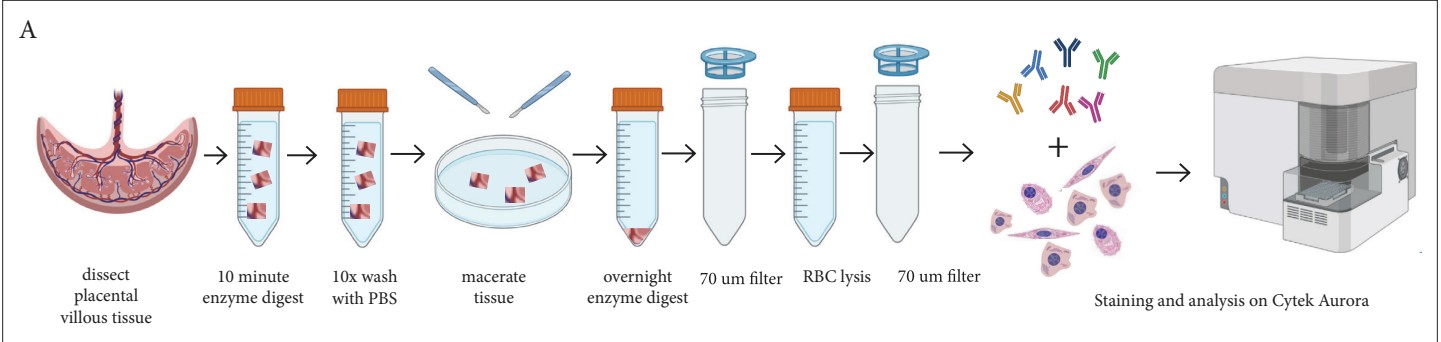

**Figure 6.** Schematic diagram demonstrating the enzymatic digestion process used to obtain a single-cell suspension of placental villous core cells for flow cytometry analysis.

## Materials and methods
### Immunohistochemistry

All placentae used in this work were collected following informed consent with approval from the Northern X Health and Disability Ethics Committee (NTX/12/06/057/AM09). First trimester placental villous tissue (7–10 weeks of gestation) was snap-frozen in OCT compound (VWR) and 5 µm cryosections of tissue were cut and fixed in ice-cold acetone for 5 minutes. Sections were blocked with Fc Blocking Reagent (Miltenyi Biotech) for 30 minutes, then incubated with primary antibodies for 1 hour (*Table 2*). β4 integrin and CD31 were visualised by their directly conjugated fluorophores (*Table 2*). CD144-BV421 was visualised by the addition of Biotin-SP AffiniPure Goat Anti-Mouse IgG (1 µg/mL) (Jackson Lab, JI115065166) for 1 hr, washing in PBS-tween, and then the addition of Streptavidin Alexa Fluor 594 (0.5µg/mL, Invitrogen) for 30 minutes. Nuclei were visualised by counterstaining with DAPI (2 µg/ml) for 5 minutes. Sections were mounted using Citifluor AF1 (Agar Scientific), and imaged on an inverted Zeiss Axioplan 2 fluorescent microscope (Zeiss).

### Isolation of placental villous core cells for ex vivo phenotyping

In order to assess placental villous core cells we modified the protocol developed by *Pelekanos et al., 2016*; *Figure 6*. In brief, villous tissue was carefully dissected away from the fetal membranes and washed thoroughly in PBS to remove maternal blood and debris. Placental explants (~1 cm$^2$) were denuded of trophoblasts (the epithelial cells that are present around the outside of placental villi) by digestion in 10 mL/1 g tissue of Enzyme Digest Solution (1 mg/mL Dispase II, 0.5 mg/mL Collagenase A and 1.5 mg/mL DNAse I (Sigma-Aldrich, USA)) in advanced-DMEM/F12 (Thermofisher) for ten minutes at 37 °C as previously described (*Boss et al., 2020*). Explants were then washed repeatedly (at least 10 times) in PBS, until cellular material is no longer released. Washed explants were then placed in a 90 mm Falcon petri dish (Corning, US) and finely macerated with two sterile disposable scalpels (Swann Morton No.20, England). The tissue was transferred to a 50 mL Falcon tube (Corning, China) with 10 mL of Enzyme Digest Solution. The Falcon tube was briefly vortexed and then placed onto a rocker at room temperature overnight. The next morning the Falcon tube containing the digested villous tissue and enzyme solution was diluted with 40 mL PBS and filtered through a 70 µm filter. The filtrate containing villous core cells was centrifuged at 220xg for five minutes, and the supernatant was removed. 10 mL of 1 x RBC Lysis Buffer (Biolegend, USA) was incubated with the cell suspension for ten minutes to remove unwanted RBCs. Remaining cells were centrifuged at 220xg for five minutes and supernatant removed prior to flow cytometry.

Large amounts of syncytiotrophoblast fragments are released during the washing stages of the first digest, and based on our experience with similar villous digest protocols (*James et al., 2015*) it is likely that the majority of the syncytiotrophoblast is removed at this stage. Remaining large fragments of syncytiotrophoblast would be removed by the 70 µm cell strainers during the digestion process, whilst smaller syncytiotrophoblast fragments would not have an intact cell membrane, and thus nuclei would stain positive for DAPI and be gated out from analysis and/or downstream cell sorting. Whilst the villous core digest protocol would also be expected to remove the vast majority of extravillous trophoblasts, to quantify the degree of potential contamination of this cell type we stained villous

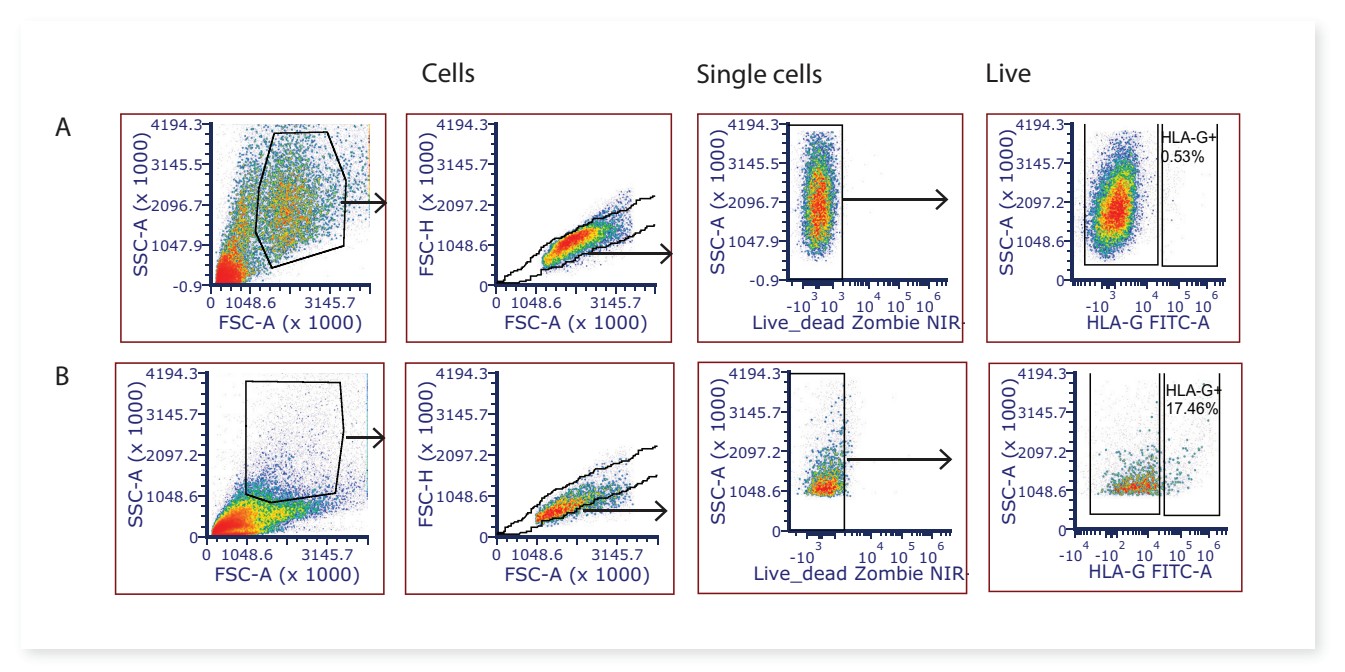

**Figure 7.** Representative 2-dimensional flow cytometry plots displaying the gating strategy used to assess the proportion of extravillous trophoblasts (HLAG +cells) in villous core digests. (**A**) Villous core cells and, (**B**) cells from the first digest washing steps known to contain extravillous trophoblasts were used as a positive control.

core cells with the extravillous trophoblast marker HLA-G (MEM-G/9)-FITC 5 µg/ml (SAB4700315, Sigma-Aldrich, NZ). A mean of 0.95% ± 0.91% of the total live cells were HLAG + extravillous trophoblasts (*Figure 7*), and are thus unlikely to impact on downstream stromal analyses. FCS files relating to quantifying extravillous trophoblasts (HLA-G + cells) contamination in villous digests can be found on FlowRepository ID FR-FCM-Z5FV.

## Multicolour panel design and optimisation

Panel One was designed for analysis on a three laser Cytek Aurora and aimed to characterise mesenchymal lineages but with specific design considerations for first trimester placental villous core cells (*Table 3*). First, antigens specific for cytotrophoblasts (an epithelial cell that is present around the outside of placental villi, identified by β4 integrin) or haematopoietic cells (CD235a, CD45) were incorporated to allow their exclusion from analysis. Then endothelial, mesenchymal, myofibroblast, and progenitor antigens were selected based on reported literature in the placental and mesenchymal fields, or included in an exploratory role based on reported antigen expression in other tissues or stem cell models (*Table 1* and *Table 3*).

The online Cytek Full Spectrum Viewer was used to identify fluorophores with distinct signatures that could be included in each panel. The panel was designed to spectrally spread out antigens with high density expression in order to prevent spectral overlap or interference. Where possible the brightest fluorophores were combined with the weakest antigens or where antigen was of special interest (ie PE) (*Ferrer-Font et al., 2020*). Conversely, where the targets of the antigen were highly expressed dim fluorophores were selected (ie CD34 PerCP).

Each antibody was titrated on first trimester placental villous core cells (*Table 3—source data 1* and *Table 3—source data 2*). For some antibodies the optimal dose was difficult to detect due to the different autofluorescence and size of cells. Therefore, forward scatter was often employed to look at different sized cells, or CD45/β4 integrin were used to exclude hematopoietic and cytotrophoblast cells that interfered with detection of optimal antibody concentration in cell populations of interest (*Table 3—source data 3*). Where required the specificity of the antibody was confirmed using immunofluorescence (*Figure 1*). Biological controls included stromal vascular fraction and peripheral blood mononuclear cells, as have been previously characterised (*Boss et al., 2020*).

**Table 3.** Panel One designed to assess villous core cells on a 3 L Cytek Aurora.
Beads = Anti Ms Ig CompBead Plus Set (BD, 560497).

| Antibody | Fluorophore | Clone | Flow cytometry Dose (µL) | Reference control | Supplier |
|---|---|---|---|---|---|
| CD55 | BB515 | IA10 | 0.6 | Villous core cells | Beckman Coulter |
| β4 integrin | FITC | 450-9D | 0.6 | Villous core cells | Thermofisher |
| CD34 | PerCP | 581 | 0.3 | Villous core cells | BD |
| CD36 | PerCpVio700 | REA760 | 0.6 | Villous core cells | Miltenyi Biotec |
| VEGFR2 | PE | 7D4-6 | 0.6 | Villous core cells | Biolegend |
| CD271 | PE-CF594 | C40-1457 | 0.6 | Stromal vascular fraction | Biolegend |
| CD142 | BB700 | HTF-1 | 1.25 | Villous core cells | BD |
| CD26 | PE/Cy5 | BA5b | 0.15 | Villous core cells | BD |
| PDPN | PE/Cy7 | NC-08 | 0.3 | Villous core cells | BD |
| CD248 | Alexa Fluor 647 | B1/35 | 0.6 | Stromal vascular fraction | BD |
| CD41 | APC | HIP8 | 0.3 | Beads | BD |
| CD90 | Alexa700 | 5E10 | 0.6 | Villous core cells | Biolegend |
| CD39 | BUV737 | TU66 | 1.25 | Beads | BD |
| ICAM1 | APC/Fire750 | HA58 | 2.5 | Stromal vascular fraction | Biolegend |
| CD144 | BV421 | 55–7 H1 | 1.25 | Villous core cells | BD |
| CD235a | Pacific Blue | H1264 | 0.3 | Villous core cells | BD |
| CD31 | BV480 | WM59 | 0.3 | Villous core cells | BD |
| CD45 | Krome-Orange | B61840 | 0.3 | Villous core cells | Beckman Coulter |
| CD146 | BV605 | PIH12 | 0.3 | Villous core cells | BD |
| CD117 | BV650 | 104D2 | 0.3 | Beads | BD |
| HLADR | BV750 | L243 | 0.3 | Villous core cells | BD |
| CD73 | BV785 | AD2 | 0.3 | Villous core cells | BD |

The online version of this article includes the following source data for table 3:

**Source data 1.** All antibodies used in Panel One were titrated on placental villous core digest cells.

**Source data 2.** Titration of additional antibodies, not contained in Panel One, required for the FACS sorting with Panel Two.

**Source data 3.** Representative images depicting how forward scatter (FSC) (A) or addition of cell-specific antibodies improved detection of appropriate doses for specific placental populations.

## Spectral flow cytometry

Freshly isolated cells were blocked with 5 µL of Human TruStain FcX (Biolegend, USA) and 5 µL of True-Stain Monocyte Blocker (Biolegend, USA) on ice for 30 minutes. An antibody master mix containing antibodies listed in *Table 3* was prepared in 5 µl of Brilliant Stain Buffer (Biosciences, US) in order to block BD Horizon Brilliant fluorescent polymer dyes from interacting with each other. The master mix was added to the cells in blocking solution to make a final volume of 50 µl and incubated on ice for 30 minutes in the dark. Stained cells were washed with 1 mL of ice cold FACS buffer (PBS, 2 mM EDTA (Thermofisher, US) and 1% FBS (Thermofisher, NZ)) and centrifuged at 220xg for five minutes at 4 °C. The supernatant was decanted and the wash was repeated. Cells were then resuspended in 100 µl of FACS buffer and kept at 4 °C. DAPI (1:5000, Akoya Biosciences) was spiked into FACS tubes directly prior to analysis on a Cytek Aurora in order to detect live/dead cells. Spectral data was processed using the Cytek SpectroFlo Software Package.

Fluorescence minus one (FMO) experiments were performed to accurately gate cell populations and to assess fluorophore spreading error where required. This was completed by creating a master

**Table 4.** Composition of Panel Two.
This panel was developed to sort CD73 + CD90 + and podoplanin + CD36 + cells from placental villous core using a BD FACS Aria II.

| Antibody | Fluorophore | Clone | Dose (µL) | Supplier |
|---|---|---|---|---|
| CD45 | FITC | HI30 | 0.3 | BD |
| PDPN | PE | NC-08 | 0.3 | Biolegend |
| CD26 | PE/Cy7 | BA5b | 0.6 | BD |
| CD271 | PE/Dazzle 594 | C40-1457 | 0.6 | BD |
| CD144 | PerCP-5.5 | 55–7 H1 | 0.6 | BD |
| CD90 | Alexa700 | 5E10 | 0.6 | Biolegend |
| CD36 | APC-Cy7 | 5–271 | 0.6 | Biolegend |
| β4 integrin | APC | 450-9D | 0.6 | Thermofisher |
| CD31 | BV480 | WM59 | 0.3 | BD |
| CD73 | BV785 | AD2 | 0.3 | BD |
| CD34 | BUV395 | 581 | 1.25 | BD |

mix containing all the antibodies in the panel minus the antibody of interest. The FMO was then compared to the full panel stain to identify whether the expression detected by that antibody is only present in the full stain, and therefore represents true positive expression. FCS files relating to the characterisation of primary cells can be found on FlowRepository ID FR-FCM-Z4TJ.

## Data analysis

Spectrally unmixed FCS files were exported from SpectroFlo and either manually analysed using 2 dimensional dot plots with FCS Express (v7) or uploaded onto Cytobank, a platform enabling datasets to be analysed by algorithms designed to assess high dimensional flow cytometry datasets https://cytobank.org/ (*Kotecha et al., 2010*). In Cytobank, debris, doublets, and dead cells were excluded by manual gating then equal sampling (200,000 villous cells from each placentae were selected, n=5). ViSNE enables visualisation of high dimensional data in two dimensions, each cell is displayed as a point on a scatter-like plot where cells with similar antigen expression will group closely and dissimilar cells will be further apart.

For further analysis of subsets of core populations, cytotrophoblasts (identified by their expression of β4 integrin) and hematopoietic cells (identified by their expression of CD45 or CD235a) were excluded by gating. Villus core subsets were then interrogated using expression heatmaps (Cytobank) and flow cytometry plots (FCS Express7). The gating strategy employed to identify populations is displayed in *Figure 3*.

## FACS sorting and in vitro culture of mesenchymal populations

In order to determine in vitro characteristics of Subset Two (CD73⁺CD90⁺) and Subset Four (podoplanin⁺CD36⁺) these populations were isolated using FACS for in vitro culture. To do this, placental villous core cells were isolated and stained with Panel Two master mix (*Table 4*) using the same methods as above. Cells were resuspended in 500 µL of Sort Buffer (10% FBS, 2 mM EDTA in PBS) and incubated on ice until immediately prior to sorting. Subset Two (CD73⁺CD90⁺) and Subset Four (podoplanin⁺CD36⁺) were sorted into 5 mL FACS tubes containing endothelial basal media (EGM-2 medium without supplements added, Lonza, USA) supplemented with 10% FBS at 4 °C degrees. To sort cells of interest, the gating strategy displayed in *Figure 3* was used to exclude debris, doublets, dead cells, trophoblasts, and endothelial cells, before gating the populations of interest. For each population 3000 cells/cm² were seeded onto 24-well plates in EGM-2. After 7 days, the in vitro phenotype of cells was determined using Panel Three (*Table 5*). In brief, TryPLE express was used to detach cells from flasks. Cells were blocked and then stained with a master mix for Panel Three (*Table 5*) as described above. Placental MSCs were isolated in EGM-2 using an explant method as described in

**Table 5.** Composition of Panel Three.

This panel was designed to assess the phenotype of placental populations after culture in vitro using a three laser Cytek Aurora.

| Antibody | Fluorophore | Clone | Dose | Supplier |
|---|---|---|---|---|
| CD34 | PerCP | 581 | 0.3 | BD |
| CD36 | PerCpVio700 | REA760 | 0.6 | Miltenyi Biotec |
| VEGFR2 | PE | 7D4-6 | 0.6 | Biolegend |
| CD271 | PE/Dazzle 594 | C40-1457 | 0.6 | Biolegend |
| CD142 | BB700 | HTF-1 | 1.25 | BD |
| CD26 | PE/Cy5 | BA5b | 0.15 | BD |
| PDPN | PE/Cy7 | NC-08 | 0.3 | BD |
| CD90 | Alexa700 | 5E10 | 0.6 | Biolegend |
| CD144 | BV421 | 55–7 H1 | 1.25 | BD |
| CD31 | BV480 | WM59 | 0.3 | BD |
| CD45 | Krome-Orange | B61840 | 0.3 | Beckman Coulter |
| CD146 | BV605 | PIH12 | 0.3 | BD |
| HLADR | BV750 | L243 | 0.3 | BD |
| CD73 | BV785 | AD2 | 0.3 | BD |

our previous work (*Boss et al., 2020*) were assessed in parallel by flow cytometry. FCS files relating to the cultured experiments can be found on FlowRepository ID FR-FCM-Z4TL.

## Immunocytochemistry

In order to determine the capacity of Subset Two (CD73+CD90+) and Subset Four (podoplanin+CD36+) to upregulate contractile markers, in a similar way to pMSCs (*Boss et al., 2020*), cells were FACS sorted (as above) and seeded into 96 well plates at 2000 cells/cm$^2$ in EGM-2, with medium replaced every 2–3 days. At day 7 medium was switched to advanced-DMEM/F12 or kept as EGM-2 for a further 7 days. At day 14 of culture, cells were fixed with methanol for 10 min, washed in PBS, then incubated in 10% normal goat serum in PBS-tween for 1 h at room temperature. Cells were incubated with primary antibodies either 5 μg/ml of anti-Calponin (Abcam-ab216651), 1 μg/ml anti-α-smooth muscle actin (αSMA) (Thermofisher Scientific, 14-9760-82) or 1 μg/ml anti-smooth muscle heavy chain 11 (Abcam-ab53219) or the relevant control antibodies rabbit IgG (Abacus, JI111036144) or mouse IgG (Abacus, JI115036146), for 1 h at room temperature. Endogenous peroxidase activity was quenched by addition of 3% $H_2O_2$ in methanol for 5 minutes. A Histostain Plus Bulk Kit (Invitrogen) with AEC chromogen (Invitrogen) was used as per the manufacturer's instructions to visualise antibody binding. Nuclei were counterstained with Gills II Haematoxylin (Sigma-Aldrich).

## Acknowledgements

The authors would like to thank all the donors of placental tissue, and staff of Epsom Day Unit, Greenlane Clinical Centre and Auckland Medical Aid Centre for their assistance in recruiting patients to the study. This work was supported by a Health Research Council of New Zealand Sir Charles Hercus Research Fellowship (J James, 16/043). A Boss was supported by a University of Auckland Doctoral Scholarship and a University of Auckland Faculty of Science PhD output award.

## Additional information

### Funding

| Funder | Grant reference number | Author |
|---|---|---|
| Health Research Council of New Zealand | 16/043 | Joanna L James |
| University of Auckland | Doctoral Scholarship | Anna Leabourn Boss |

The funders had no role in study design, data collection and interpretation, or the decision to submit the work for publication.

### Author contributions

Anna Leabourn Boss, Conceptualization, Data curation, Formal analysis, Investigation, Visualization, Methodology, Writing - original draft; Tanvi Damani, Investigation; Tayla J Wickman, Data curation, Formal analysis; Larry W Chamley, Writing – review and editing; Joanna L James, Conceptualization, Resources, Supervision, Funding acquisition, Methodology, Project administration, Writing – review and editing; Anna ES Brooks, Conceptualization, Resources, Software, Supervision, Funding acquisition, Methodology, Writing – review and editing

### Author ORCIDs

Anna Leabourn Boss (i) http://orcid.org/0000-0002-1943-4162
Anna ES Brooks (i) http://orcid.org/0000-0003-3551-6982

### Ethics

Human subjects: Placentae were collected following informed consent with approval from the Northern X Health and Disability Ethics Committee (NTX/12/06/057/AM09).

### Decision letter and Author response

Decision letter https://doi.org/10.7554/eLife.76622.sa1
Author response https://doi.org/10.7554/eLife.76622.sa2

## Additional files

### Supplementary files

• Transparent reporting form

### Data availability

FCS data files have been provided for flow cytometry presented in Figures 2-3 following the link: http://flowrepository.org/public_experiment_representations/FR-FCM-Z4TJ Figure 4: http://flowrepository.org/public_experiment_representations/FR-FCM-Z4TL Figure 7: http://flowrepository.org/public_experiment_representations/FR-FCM-Z5FV.

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
