## [Editor Report]

Placental mesenchymal stromal cells (pMSCs) are of interest in therapeutic applications. These cells are typically generated by culturing cells isolated from the villous core of the placenta. However, the villous core is comprised of multiple cell types and the heterogeneity of these cells is often not considered. Consequently, the origin of pMSCs under commonly used culture conditions remains unclear. In the present study the authors have used sophisticated flow cytometry analysis to characterize the heterogeneity of subtypes in the placental villus core in the first trimester of gestation and present convincing data that a specific subpopulation identified likely corresponds to pMSCs in culture generated using standard isolation protocols. This study will be valuable to scientists investigating the use of placental mesenchymal stromal cells in a therapeutic context.

---

## [Decision Letter]

**Decision letter after peer review:**

Thank you for resubmitting your work entitled "The origins of placental mesenchymal stromal cells: Full spectrum flow cytometry reveals mesenchymal heterogeneity in first trimester placentae, and phenotypic convergence in culture" for further consideration by *eLife*. Your revised article has been reviewed by three peer reviewers, one of whom is a member of our Board of Reviewing Editors, and the evaluation has been overseen by Ricardo Azziz (Senior Editor).

The manuscript has been improved but there are some remaining issues that need to be addressed, as outlined below:

*Reviewer #1 (Recommendations for the authors):*

The authors need to characterize what is in the isolated suspension, and the supporting experiments and pieces of evidence, authors excluded the trophoblast population in the culture. For example, hCG secretion and syncytin expression in the culture population.

How the various PMS are derived in the culture is not clear, more importantly, the functional characterization of each population of subsets is missing in the manuscript.

The limitations of using this phenotype in the therapeutic application need to be emphasized in the manuscript,

Is there any paracrine communication that exists between PMSC and cyto's needs to be addressed? What are the defects arises in terms of this PMSC and various pregnancy complications that need to be addressed?

This manuscript is well written, and the authors well executed the experiments.

*Reviewer #2 (Recommendations for the authors):*

Well written manuscript.

*Reviewer #3 (Recommendations for the authors):*

Placental mesenchymal stromal cells (pMSCs) are of interest in therapeutic applications. These cells are typically generated by culturing cells isolated from the villous core of the placenta. However, the villous core is comprised of multiple cell types. Consequently, the origin of pMSCs under commonly used culture conditions remains unclear. In this paper, Boss et al., used flow cytometry analysis using an extensive panel of 23 cell surface markers to characterize the heterogeneity of cell types in the villous core of the first trimester placenta. They uncovered four cell populations of interest: (1) CD31+ CD34+ (2) CD73+ CD90+ (3) CD146+ CD271+ and (4) podoplanin+ CD36+. They chose to focus on two cell populations – (2) CD73+ CD90+ because these are the only cell types that express the generally accepted marker criteria for MSCs, and (4) podoplanin+ CD36+ because these express markers of proliferative/migratory cells that are associated with pMSCs.

Subsequently, the two selected cell subpopulations were cultured in pMSC medium and compared with pMSCs obtained using standard protocols (i.e. without accounting for heterogeneity of the input cells). Interestingly, the two selected subpopulations converged to an extent in terms of marker expression. Nevertheless, careful comparison revealed that pMSCs derived using standard conditions are closer to podoplanin+ CD36+ (4) cells.

Strengths:

Currently, the specific cell population(s) from primary tissue that get expanded or stabilized pMSC cell culture remain unknown or poorly characterized. Uncertainty arises from (i) heterogeneity in the source cells that are used to originate the pMSC culture and (ii) the change in cell phenotype that occurs upon culture, dependent on specific medium composition used. This paper is arguably one of the few – if not the only one – that characterizes the heterogeneity of cells used to originate pMSC cultures. In this context, the results presented in the paper are an important contribution to the literature. The methods employed by the authors are rigorous and the conclusions drawn are largely consistent with experimental results reported.

Weaknesses:

The authors use flow cytometry to characterize biomarker expression in the cells of the villous core. However, while highly relevant, such characterization is one dimensional. Not including data on characterization of the functional differences of the two subpopulations focused on in terms of differentiation potential or transcriptome differences (using bulk RNAseq) between the subpopulations is a significant missed opportunity.

Figure 4 can perhaps be presented in a way as to capture the rich information present in the dataset. For instance, perhaps fluorescence intensities can be included as violin plots rather than just using percent positive cells. Similarly, maybe alternative data visualization can be used to comprehensively depict overlap or differences between the two subpopulations at day 0 and day 7 and pMSCs from explant cultures.

The primary data for flow cytometry should be made available.

---

## [Author Response]

Reviewer #1 (Recommendations for the authors):The authors need to characterize what is in the isolated suspension, and the supporting experiments and pieces of evidence, authors excluded the trophoblast population in the culture. For example, hCG secretion and syncytin expression in the culture population.

The reviewer raises a valid point that it is important to know what is not in the isolated suspension, and in particular to ensure that extravillous trophoblasts and syncytiotrophoblast are removed. The initial digest step in the methods used in this work was modified from that used by James et al., (PMID 26248480) to isolate pure populations of extravillous trophoblast (and subsequently cytotrophoblast). The first digestion step in both protocols involves enzymatic digestion, followed by extensive washing in PBS which releases both syncytiotrophoblast fragments and HLA-G+ extravillous trophoblasts. Therefore, we did not expect significant extravillous trophoblast contamination of our samples. However, in response to the reviewers comment we have now used the extravillous trophoblast marker HLA-G to quantify any extravillous trophoblast contamination within our villous core digests. This demonstrated that <2% of cells are extravillous trophoblasts. Furthermore, prior data has also demonstrated that extravillous trophoblasts do not express mesenchymal markers (CD90, CD36, CD271, CD26 or PDPN) used to gate populations of interest in this manuscript (Suryawanshi et al., 2018). Thus, together we are confident that this low level of contamination would not impact the data.

As the syncytiotrophoblast is a large single multinucleated cell, fragments can vary widely in size, and ensuring the syncytiotrophoblast was adequately removed was as such important in developing these methods as larger multinucleated cells had the potential to block the cell sorter. In this work large fragments are removed via the use of 70um cell strainers throughout the digestion process, whilst smaller syncytiotrophoblast fragments would not have an intact cell membrane, and thus nuclei would stain positive for DAPI and gated out of the analysis and/or downstream cell sorting.

We have now added to the methods section of the manuscript the new experimental data quantifying HLA-G+ extravillous trophoblast contamination in placental villous samples (Figure 7) and outlined why syncytiotrophoblast fragments would be excluded from analysis (lines 446-453).

How the various PMS are derived in the culture is not clear, more importantly, the functional characterization of each population of subsets is missing in the manuscript.

We agree that functional experiments are an important future downstream step from this work and have now further discussed this at line 380-383. The work in the current manuscript focussed on characterising mesenchymal phenotypic heterogeneity prior to culture and how culturing in standard medium could influence this, and in doing so has provided a comprehensive analysis with novel insight. Whist the degree to which the phenotypic convergence of distinct in vivo stromal populations in culture results in their functional convergence remains unclear, in order to assess this properly (or to gain better insight into their in vivo function) it would first be important to identify specific culture conditions that can maintain the distinct in vivo phenotypes of each stromal population. Such functional assays would also be dependent on the specific downstream context for use of the cells, and as such the experiments required would be substantial and we believe that this is beyond the scope of this paper.

The limitations of using this phenotype in the therapeutic application need to be emphasized in the manuscript,

At line 366-369 we discuss the concept that different subsets identified in this work may be better suited for different applications. Furthermore, we have provided several examples of the potential therapeutic consequences of using the different subsets identified in this work 373-377. Whilst ideal for identifying novel early progenitor populations and understanding the role of progenitors in early placental development, a key limitation in the therapeutic use of these cells is their derivation from first trimester tissue, and we have now acknowledged this and the need for future work to examine the presence of counterparts in term placentae at line 381-383.

Is there any paracrine communication that exists between PMSC and cyto's needs to be addressed?

Yes, there is evidence of paracrine relationships between trophoblasts and mesenchymal populations in the placenta, and this is an interesting topic that we have covered as part of a recently accepted review (James et al., 2022, Cellular and Molecular Life Sciences, In Press). However, this is largely thought to work in the direction of the mesenchyme influencing cytotrophoblast function and extravillous trophoblast differentiation, rather than the other way around, and is not directly relevant to this manuscript.

What are the defects arises in terms of this PMSC and various pregnancy complications that need to be addressed?

Despite gross anatomical differences in the placental stroma and vasculature, it has only been very recently that differences in pMSCs in pregnancy complications have been assessed by ourselves and others. We have now included reference to the potential role of MSCs in pregnancy pathologies, and highlighted the potential future use of this panels to better understand these disorders at line 393-397.

Reviewer #3 (Recommendations for the authors):Placental mesenchymal stromal cells (pMSCs) are of interest in therapeutic applications. These cells are typically generated by culturing cells isolated from the villous core of the placenta. However, the villous core is comprised of multiple cell types. Consequently, the origin of pMSCs under commonly used culture conditions remains unclear. In this paper, Boss et al., used flow cytometry analysis using an extensive panel of 23 cell surface markers to characterize the heterogeneity of cell types in the villous core of the first trimester placenta. They uncovered four cell populations of interest: (1) CD31+ CD34+ (2) CD73+ CD90+ (3) CD146+ CD271+ and (4) podoplanin+ CD36+. They chose to focus on two cell populations – (2) CD73+ CD90+ because these are the only cell types that express the generally accepted marker criteria for MSCs, and (4) podoplanin+ CD36+ because these express markers of proliferative/migratory cells that are associated with pMSCs.Subsequently, the two selected cell subpopulations were cultured in pMSC medium and compared with pMSCs obtained using standard protocols (i.e. without accounting for heterogeneity of the input cells). Interestingly, the two selected subpopulations converged to an extent in terms of marker expression. Nevertheless, careful comparison revealed that pMSCs derived using standard conditions are closer to podoplanin+ CD36+ (4) cells.Strengths:Currently, the specific cell population(s) from primary tissue that get expanded or stabilized pMSC cell culture remain unknown or poorly characterized. Uncertainty arises from (i) heterogeneity in the source cells that are used to originate the pMSC culture and (ii) the change in cell phenotype that occurs upon culture, dependent on specific medium composition used. This paper is arguably one of the few – if not the only one – that characterizes the heterogeneity of cells used to originate pMSC cultures. In this context, the results presented in the paper are an important contribution to the literature. The methods employed by the authors are rigorous and the conclusions drawn are largely consistent with experimental results reported.Weaknesses:The authors use flow cytometry to characterize biomarker expression in the cells of the villous core. However, while highly relevant, such characterization is one dimensional. Not including data on characterization of the functional differences of the two subpopulations focused on in terms of differentiation potential or transcriptome differences (using bulk RNAseq) between the subpopulations is a significant missed opportunity.

We agree that using RNAseq to better understand the stromal subsets identified in this work could generate a wealth of data, but do not currently have the funding available to undertake this. However, we have included discussion of a recent manuscript that interrogated first trimester placental tissue using single-cell RNAseq and found mesenchymal heterogeneity that aligns with our hypothesised mesenchymal populations (lines 265, 315, 323). Our work using a high-dimensional flow cytometry panel is novel in that it can be used to select and sort cells (as done in this work), and measures protein expression which can vary from RNA and is likely more relevant to cell functionality (Buccitelli and Selbach, 2020, PMID: 32709985).

Figure 4 can perhaps be presented in a way as to capture the rich information present in the dataset. For instance, perhaps fluorescence intensities can be included as violin plots rather than just using percent positive cells. Similarly, maybe alternative data visualization can be used to comprehensively depict overlap or differences between the two subpopulations at day 0 and day 7 and pMSCs from explant cultures.

We have amended the graphs in figure 4 so that each sample can be visualised rather than creating violin plots that demonstrate intensity levels. We are identifying switching on or off of marker expression. Small shifts in fluorescence intensity of these markers might not represent functional differences. Due to the inherent dramatic changes in cellular characteristics (e.g. autofluorescence) with culturing cells it is difficult to compare primary (day 0) with cultured (day 7) cells and this is why we have not directly compared populations between timepoints. However, we have created figure 4- supplement figure 1 and 2 (previously supplementary figure 4 and 5) in order to allow easier comparison of marker expression between populations at day 7. The cultured mesenchymal cells demonstrate a relatively homogenous phenotype therefore this was not displayed as 2-dimensional flow cytometry plots in the main manuscript as these were not informative. However, we have now included representative 2-dimensional flow cytometry plots, post culture, of CD36+podoplanin+ cells, CD73+CD90+ cells and explant pMSCs (Figure 4- Supplement Figure 1).

The primary data for flow cytometry should be made available.

Primary data has been uploaded onto FlowRepository. The reference IDs for this data have been added to the methods lines 434, 491 and 522.